# Potential of Resveratrol to Combine with Hydrogel for Photodynamic Therapy against Bacteria and Cancer—A Review

**DOI:** 10.3390/biomedicines12092095

**Published:** 2024-09-13

**Authors:** Siu Kan Law, Cris Wai Ching Liu, Christy Wing Sum Tong, Dawn Ching Tung Au

**Affiliations:** Department of Food and Health Sciences, The Technological and Higher Education Institute of Hong Kong, Tsing Yi, New Territories, Hong Kong, China; criswcliu@thei.edu.hk (C.W.C.L.); christytong@thei.edu.hk (C.W.S.T.)

**Keywords:** resveratrol, hydrogel, photodynamic therapy, bacteria, cancer

## Abstract

Bacterial infections and cancers are important issues in public health around the world. Currently, Western medicine is the most suitable approach when dealing with these issues. “Antibiotics” and “Corticosteroids” are the Western medicines used for bacterial infection. “Chemotherapy drugs”, “surgery”, and “radiotherapy” are common techniques used to treat cancer. These are conventional treatments with many side effects. PDT is a non-invasive and effective therapy for bacterial infection and cancer diseases. Methods: Nine electronic databases, namely WanFang Data, PubMed, Science Direct, Scopus, Web of Science, Springer Link, SciFinder, and China National Knowledge Infrastructure (CNKI), were searched to conduct this literature review, without any regard to language constraints. Studies focusing on the photodynamic actions of hydrogel and Resveratrol were included that evaluated the effect of PDT against bacteria and cancer. All eligible studies were analyzed and summarized in this review. Results: Resveratrol has antibacterial and anticancer effects. It can also act as PS in PDT or adjuvant but has some limitations. This is much better when combined with a hydrogel to enhance the effectiveness of PDT in the fight against bacteria and cancer. Conclusions: Resveratrol combined with hydrogel is possible for PDT treatment in bacteria and cancer. They are compatible and reinforce each other to increase the effectiveness of PDT. However, much more work is required, such as cytotoxicity safety assessments of the human body and further enhancing the effectiveness of PDT in different environments for future investigations.

## 1. Introduction

Bacterial infection and cancer disease are issues involved in public health around the world. Currently, Western medicine is the common approach to managing these conditions. 

Antibiotics and corticosteroids are medicines that are used to fight bacterial infections. They either kill or stop bacterial growth by affecting self-reproduction and preventing mutation, gene recombination, and transfer. This allows the body’s natural defenses to eliminate the pathogens [1]. The negative impacts of antibiotics are increasing morbidity and mortality, leading to antimicrobial resistance (AMR), which poses a major threat to human health in the world [2].

Chemotherapy drugs, surgery, and radiotherapy are the general techniques used to fight against cancer, but these have side effects on the human body [3]. Therefore, photodynamic therapy (PDT) has become more popular in medical usage.

PDT is a non-invasive therapy, which is different from traditional surgery. This is a selective and repeatable treatment, unlike radiation therapy, which is harmful to normal cells or tissues. Compared with antibiotics and corticosteroid chemotherapy, PDT is more effective and has fewer side effects [4].

Resveratrol is a natural product that has been extracted from Chinese herbal plants [5], and it has been applied for the treatment of different diseases, such as bacterial infections and cancer. It also can act as a photosensitizer (PS) in PDT therapy or adjuvant. If it is a PS, a photo-induced chemical reaction of *trans*-Resveratrol occurs to generate singlet oxygen (^1^O_2_) during the PDT process [6].

PDT has been used for a long time in the treatment of bacterial infections and cancer. Antibacterial photodynamic therapy (aPDT) has the potential to kill multidrug-resistant pathogenic bacteria, such as Gram-positive bacteria (*Staphylococcus aureus* and *Enteroccoccus fecalis*) and Gram-negative bacteria (*Escherichia coli*, *Proteus mirabilis*, and *Pseudomonas aeruginosa*). It has a low tendency to induce drug resistance, and bacteria rapidly develop against traditional antibiotic therapy [7]. PDT enables the treatment of multifocal disease with the least amount of tissue damage in terms of cancer [8]. The mechanisms of PDT, either in bacterial infection or cancer, are quite similar and are involved in Types I and II to generate the singlet oxygen (^1^O_2_) and reactive oxygen species (ROS) in the electron-transfer reaction.

Resveratrol can act as a natural PS, but it has some limitations for PDT, for example, its short wavelength and poor bioavailability. It is usually combined with hydrogel to enhance the function of PDT. Resveratrol and hydrogel are complementary during the PDT process in terms of maximizing effectiveness. Recently, Gan and Liu et al. have published a similar review of hydrogel-based phototherapy and drug-delivery systems [9,10], but these do not describe the natural product Resveratrol.

Hence, this review article is divided into five parts, namely (1) the definition of hydrogel, (2) the theory and mechanisms of PDT, (3) natural PS, especially in the case of Resveratrol, (4) the photodynamic action of hydrogel, and (5) the possibility of combining hydrogel with Resveratrol for PDT against bacteria and cancers.

### 1.1. Hydrogels

Hydrogels are three-dimensional (3D) network structures with crosslinked polymer chains that form ionic or covalent bonds for linking one polymer chain to another [11]. Glutaraldehyde [12,13] is the most common compound for crosslinking a hydrogel, and other crosslinking compounds include Formaldehyde [14], Epoxy [15], etc. (Figure 1). The crosslinking changes a liquid polymer into a solid or gel by restricting its ability to move. Since the crosslinking structure of a hydrogel can absorb relatively large amounts of water, its properties are soft and resemble living tissues in humans [16].

Hydrogels are porous, with spaces available between adjacent crosslinking structures in the polymer network [17]. There are some development theories for porous hydrogels, including the kinetics of swelling, equilibrium swelling, the structure–stiffness relationship, and solute diffusion in dense hydrogels.

The swelling of a hydrogel is the kinetic process of coupling mass transport and mechanical deformation that includes linear and non-linear poroelasticity. These theories are consistent within the linear regime under small perturbations from an isotropically swollen hydrogel state, such as a change in the volume of a hydrogel as it absorbs a compatible solvent or water content [18]. Hydrogel also causes equilibrium swelling depending on the pH of an environment. It acts as a polyampholyte polyelectrolyte when immersed in an ionic solution, producing hydrogen and hydroxide ions in a reaction [19].

This is the soft or weak material with a structure–stiffness relationship. Water content and mechanical properties of the hydrogel are adjusted over a wide range through the dissociation and reformation of hydrogen bonds by solvent exchange and the heating process [20]. Diffusion of solutes within the hydrogel occurs through nano-to-microscopic open spaces or dynamic free volumes between the aqueous solution and the liquid-filled polymer fibers, which includes sub-nanometer-scale cavities between molecules [21].

### 1.2. Examples of Antibacterial and Anticancer Applications

Hydrogels are naturally derived from alginate and chitosan or synthetic modification from polyacrylamide. Fasiku et al. reported a chitosan-based hydrogel for dual delivery with hydrogen peroxide of antimicrobial peptide against bacterial methicillin-resistant *Staphylococcus aureus* biofilm-infected wounds that were prepared through the Michael addition technique (Figure 2) [22]. In 2022, Abbasalizadeh et al. developed a curcumin-chrysin-alginate-chitosan hydrogel that was prepared through the ionic gelation mechanism utilizing CaCl_2_ to treat breast (T47D) and lung cancers (A549) (Figure 3) [23]. Lu et al. identified the polyacrylamides hydrogel causing *Staphylococcus aureus* and *Escherichia coli* differentiation upon visible light irradiation (Figure 4) [24]. Andrade et al. indicated that stimuli-responsive hydrogel was able to change its physical state from liquid to gel according to external factors such as temperature, pH, light, ionic strength, and magnetic field for cancer treatment (Figure 5) [25].

### 1.3. Drug-Delivery System

There are some development theories of hydrogels to the drug-delivery approach. First, hydrogel establishes scaffolds to address easily inadequate local drug availability and delivery sites. These have a highly hydrated mesh network formed from natural, synthetic, or semi-synthetic polymers with physically or covalently crosslinked (Table 1) [26]. They provide high biocompatibility [27], drug protection [28], spatiotemporal control of the drug release [29], and physicochemical tailorability [30].

Biocompatibility is the ability of a biomaterial (e.g., hydrogel) to perform with an appropriate host response in the specific application [27]. Hydrogels also serve as a platform on which various physiochemical interactions with the encapsulated drugs occur to control drug release [28]. This provides a wide range of drug–carrier interface modifications to maximize drug-delivery needs through physical (electric field, magnetic field, light), mechanical (ultrasound, mechanical strain), or chemical (pH, redox gradient, enzymes), and without side effects [29].

Moreover, the physicochemical tailorability sparks through the inherent bioactivity of gelatin in tissue engineering. Spatial and temporal control of hydrogel crosslinking and its rheological behavior enables deposition via a variety of manufacturing-related techniques. Hydrogels can tailor to the remodeling rate of the target tissue to produce viable well-defined 3D structures or long-term cellular performance in non-biofabricated structures [30]. Therefore, hydrogel is a suitable material as a drug-delivery system applied in PDT.

### 1.4. Reasons for Hydrogel Use in PDT

In general, hydrogel has several competitive advantages, including good biocompatibility and low cytotoxicity, enhanced antitumor effect, reduced toxic and side effects, as well as maintaining a high concentration and archive the selective of a drug during the PDT process. These characteristics of hydrogel overcome the limitations of PDT and enhance its effectiveness, which has been discussed in drug-delivery systems. Previous studies have investigated a novel hydrogel shell on cancer cells that were prepared through in situ photopolymerization of polyethyleneglycol diacrylate (PEGDA) using methylene blue (MB)-sensitized mesoporous titanium oxide (TiO_2_) nanocrystal for enhancing the effectiveness of PDT. TiO_2_ was the PS and activated the formation of hydrogel, which protected the MB and acted as a significant photosensitive additive to improve the treatment of PDT. MB would be eliminated and inactivated after undergoing the PDT process [31]. Thus, there is a relationship between the principle of PDT and hydrogel.

## 2. Principle of PDT

The basic principle of PDT is a dynamic interaction between the photosensitizer (PS) and light with a specific wavelength that generates the reactive oxygen species (ROS), such as singlet oxygen (^1^O_2_) and superoxide anion (O_2_^−^) that causes irreversible oxidative damage to the cell membrane and DNA, leading to the selective destruction of target tissue, bacteria, and cancer [32].

Conventional PDT with the advantages of direct targeting, low invasiveness, remarkable curative, and fewer side effects [33]. However, it has limited or weak light-penetration depth, a short release distance, and a lifetime of ROS to fight against bacteria [34]. This also exists alongside poor selectivity and oxygen dependence for the therapeutic effectiveness of solid tumors [35]. Table 2 summarizes the representative cases of recent progress in PDT against bacteria and cancer over five years.

### 2.1. Mechanisms

In general, PDT causes the activation of PS through visible or ultraviolet light, which generates the ^1^O_2_ and free radicals. The electrons of the PS are promoted from the ground to an excited state by the light activation. These excited singlet-state electrons from the PS are unstable and release the excess energy backing to the ground state by emitting fluorescence. Its excited singlet-state electrons can also change and shift to the triplet state through the intersystem crossing system [42].

There are two types of pathways for the triplet state of PS that react with the substrate [43]. Type-I and II pathways involve the electron transfer from triplet state PS. The difference between these two pathways is the generation of final products. Type-I pathway consists of free radicals, and these interact with oxygen at a molecular level to form ROS, such as superoxide (O_2_^•−^), hydroxyl radicals (•OH), and hydrogen peroxide (H_2_O_2_). Type II pathway contains ^1^O_2_ only generated by the electron transfer between the excited PS and the ground-state molecular oxygen [44]. Type-I pathway operates in an oxygen (O_2_)-independent manner, and the type II pathway relies on the presence of molecular oxygen (Figure 6).

These Type-I and II pathways of antimicrobial photodynamic therapy (aPDT) are employed for killing Gram-positive bacteria (*Staphylococcus aureus*, *Enteroccoccus fecalis*) and Gram-negative bacteria (*Escherichia coli*, *Proteus mirabilis*, and *Pseudomonas aeruginosa*) bacteria [42], which causes irreversible oxidative damage to the cell membrane and DNA, leading to bacterial cell death [45]. Both Type-I and II pathways are also applied in the treatment of cancers that can trigger different cell death mechanisms and are directly cytotoxic to the cancer cells resulting in apoptosis or necrosis [46]. The percentage or ratio of the final products for Type-I and II pathways depends on the nature or type of PS.

### 2.2. Natural Photosensitizer (PS)

Typically, PS molecules absorb light with a suitable wavelength that initiates the activation processes for the selective destruction of cancer cells [47]. Many natural PS from traditional Chinese medicinal plants are applied in PDT, including Pheophorbide a (Pa) [48], Hypocrellin B (HB) [49], and Curcumin (Cur) (Figure 7) [50], since these are phytochemical compounds with less- or non-toxic side effects [51].

They usually act as antibacterial and anticancer agents of natural products origin [52]. PS can selectively target microbes and leave out normal tissue, making it more efficient against the bacterial infection site when as an antibacterial agent. If the PS is an anticancer agent, it reduces toxicity to healthy tissues and has a lower incidence of side effects [53]. Some representative examples of natural PS for PDT against bacteria and cancer are shown in Table 3. Resveratrol as a PS will be discussed in the next section.

#### 2.2.1. Resveratrol (3,5,4′-Trihydroxystilbene) (RSV)

This is a plant compound with a natural polyphenol stilbene structure that is isolated from the root of *Veratrum grandiflorum* [59] or *darakchasava*, or *manakka* for medicinal purposes [60]. Its molecular weight of 228.25 g/mol consists of two phenolic rings formed by a double styrene bond. Thus, there are two isomeric *cis*- and *trans*-forms (Figure 8).

The *cis*- and *trans*- forms of Resveratrol can co-exist in a variety of fruits, including grapes (Vitaceae) [61] and blueberries (*Vaccinium* spp.), blackberries (*Morus* spp.), and peanuts (*Arachis hypogaea*) [62,63], and red wine. *Trans*-Resveratrol is the predominant form and is more stable than *cis*-Resveratrol in nature [64,65]. It is a bioactive compound with a wide range of pharmacology properties, e.g., immunomodulatory [66], glucose and lipid regulatory [67], neuroprotective [68], cardiovascular protective effects [69], antioxidant [70], anti-inflammatory [71], antibacterial [72,73,74,75,76,77], and anticancer (Table 4) [78,79,80,81,82,83,84,85,86,87].

Table 4 shows the investigation of an antibacterial and anticancer effect on Resveratrol. The antibacterial mechanisms of Resveratrol consist of DNA damage [88], cell division impairment [89], oxidative membrane damage [90], and metabolic enzyme inhibition [91].

Resveratrol increases the expression of genes from a SOS–stress regulon, which causes DNA fragmentation, cell cycle arrest, and DNA impairment in bacteria, such as *Escherichia coli* [72,88]. It can also affect the prokaryotic cell division protein (FtsZ) and GTPase activity. This protein is highly conserved throughout eubacteria, which may cause cell division impairment in bacteria [92].

The effect of Resveratrol on *E. coli* is not through diffusible ROS. It is mediated site-specifically as the primary event for oxidative damage of the cell membrane [90]. It also inhibits ATPase and the synthesis of ATP bound in a hydrophobic pocket between the C-terminal tip of the gamma subunit and the hydrophobic interior of the surrounding torus, a region critical for gamma-subunit rotation [91].

Resveratrol has several anticancer mechanisms, including initiating the apoptosis of mitochondria in the cytoplasm, enhancing oxidative stress, and interfering with energy metabolism in cancer cells. The generation of ROS causes cancer cell damage and death [93].

Mitochondria is an intrinsic apoptotic pathway causing stimulation through the activation of the proapoptotic Bcl-2 family (Bak, Bax). By suppressing the anti-apoptotic proteins Bcl-xL, Bcl-2, and Mcl-1, it diminished the potential of mitochondrial outer membrane permeability (MOMP). The cytochrome c then binds to this cytosolic apoptotic protease activating factor-1 (Apaf-1) [94], which recruits the initiator procaspase-9 activating downstream executor caspase-3, -6, and -7 for cleavage of cytoplasm leading to the cell apoptosis [95].

The intervention is an extrinsic apoptotic pathway involving tumor necrosis factor (TNF) to interact with cell surface receptors such as TRAILR1 (DR4)-TRAIL, TRAILR2 (DR5)-TRAIL, TNFR1-TNFα and FAS (CD95, APO-1)-FasL [96]. It causes apoptosis by the activation of caspase-3, -6, and -7, which leads to the disassembly of cell survival and death [97].

Meanwhile, Resveratrol inhibits the neurogenic locus notch homolog protein 1 to increase the generation of ROS. These signal transductions upregulate the phosphatase tensin homolog and down-regulate AKT Serine/Threonine Kinase 1. As a result, caspase-3 cleavage by decreasing phospho-Akt induced cell death [98]. Growing evidence has indicated that *trans*-Resveratrol has anticancer properties. It is a PS applied to the PDT to enhance its functions against bacteria and cancer.

#### 2.2.2. Photodynamic Action of Resveratrol

*Trans*-Resveratrol has been reported to have significant photodynamic activity. The generation of singlet oxygen by *trans*-Resveratrol after light irradiation is used as PS for the photooxidation of ergosterol via a [4+2] cycloaddition (Figure 9) [99].

Singlet oxygen is a ROS generated by ground-state (triplet) oxygen excitation. This is the energy transfer from a photoexcited PS in the PDT process [100]. Resveratrol quinone is the main source for quenching ^1^O_2_. Its mechanism is based on the resorcinol moiety and the carbon–carbon double bond [101]. It has an excellent triplet-quenching activity in a lower concentration range of 52 μM [102].

Regarding *trans*-Resveratrol with antibacterial and anticancer functions, the ROS causes the inactivation of bacteria, which oxidizes proteins or lipids, leading to bacteria death [103]. It also inhibits the proliferation of cancer cells due to increased metabolic rate, gene mutation, and relative hypoxia [104], as well as suppressing cell growth accompanied by apoptosis evoked through increased intracellular ROS levels in mitochondria [105]. Some examples of Resveratrol as a PS for PDT against bacteria and cancer are shown in Table 5.

## 3. Photodynamic Action of Hydrogel

Hydrogels are an effective way to release drugs and antibacterial agents that greatly improve the utilization of antibacterial agents in bacteria and reduce the toxic effects on normal cells [111,112]. It is also a potent novel medication that targets cancer cells and reduces damage to normal tissues during PDT treatment. They act as a powerful drug-delivery capacity for a precisely controlled drug release because hydrogels can be loaded for many therapeutic agents, such as chemotherapeutic agents, radionuclides, and immunosuppressants, to induce a cascade of multiple therapeutic modalities. This can operate as a carrier and release system of PS for PDT, which also enhances the light-penetration depth and enables an integrated sequence of cancer treatment [113].

Hydrogels have a variety of sizes, from macrogels, microgels (0.5 to 10 μm), and nanogels (less than 200 nm) during the PDT treatment, which allows precise access to the cancer site and provides continuous as well as controlled drug delivery for the selectivity of tumors. This also minimizes the usage of drugs and reduces systemic toxicity to the corresponding tissues [114]. The hydrogel system is highly dependent on internal and external environmental stimulations, such as pH, temperature, redox potential, and reactive oxygen concentration. Meanwhile, oxygen dependence is an important factor for PDT treatment. Thus, the hydrogel system can target cancer cells and release drugs to improve the therapeutic effectiveness of solid tumors [115]. Some examples of hydrogel for PDT against bacteria and cancer (Table 6).

PDT can lead to additional crosslinking within the hydrogel network, which reacts with polymer chains, altering the mechanical properties and swelling behavior of the hydrogel. The incorporation of photosensitizers into hydrogels greatly enhances their localized concentration and biocompatibility. It prolongs their residence time to generate ROS against bacteria and cancer during the PDT process [9]. However, PDT may also cause the degradation of hydrogel to release a drug (e.g., Resveratrol) since ROS breaks down polymer chains at the same time, resulting in fragmentation and affecting overall stability. The drug-delivery system is the most important point for the hydrogel applications on antibacterial and anticancer functions [116].

**Table 6 biomedicines-12-02095-t006:** Photodynamic action of hydrogel against bacteria and cancer.

Bacteria					
	Study	Photosensitizer and Dosage	Usage of Light and Energy (J)	Consequences	Reference
1	Photosensitizer-loaded hydrogels for photodynamic inactivation of multiresistant bacteria in wounds	326 μM of tetrakis(1 methylpyridinium-4-yl)porphyrin p-toluenesulfonate (TMPyP) and 242 μM of tetrahydroporphyrin—p toluenesulfonate (THPTS) on *Enterococcus faecium*, *Staphylococcus aureus*, *Klebsiella pneumonia*, *Acinetobacter baumannii*, *Pseudomonas aeruginosa*, *Escherichia coli*, and *Achromobacter xylosoxidans*.	Irradiated with red light at 760 nm in 18 W/cm^2^, and fluence 20 J/cm^2^ for 36 to 90 min.	TMPyP-loaded hydrogels were more effective than those loaded with THPTS, which displayed effectivity against all investigated bacteria strains, improving the treatment of wounds infected with problematic bacterial pathogens.	[117]
2	An injectable dipeptide–fullerene supramolecular hydrogel for photodynamic antibacterial therapy	200 μM of dipeptide–fullerene supramolecular hydrogel on *Staphylococcus aureus* in wound healing.	Irradiated with white light at 400 nm in 0.1 W/cm^2^, and fluence 10 J/cm^2^ for 10 min.	Peptide fullerene hydrogels inhibited multi-antibiotic-resistant *Staphylococcus aureus* and promote wound healing.	[118]
3	Optimization of hydrogel containing toluidine blue O for photodynamic therapy in treating acne	0.1 μM of toluidine blue O on *Propionibacterium acnes*, *Staphylococcus aureus*, and *Escherichia coli*.	Irradiated with red light at 630 nm in 0.4 W/cm^2^, and fluence 13 J/cm^2^ for 15 min.	Toluidine blue O hydrogel for PDT showed effective antibacterial activity for *Propionibacterium acnes*, *Staphylococcus aureus*, and *Escherichia coli*.	[119]
**Cancer**					
	**Study**	**Photosensitizer and** **Dosage**	**Usage of Light and Energy (J)**	**Consequences**	**Reference**
1	Synthesis and Characterization of Temperature-sensitive and Chemically Crosslinked Poly(N-isopropylacrylamide)/Photosensitizer Hydrogels for Applications in Photodynamic Therapy	90 μM of Pheophorbide a-poly(N-isopropylacrylamide) nanohydrogel on Human Colorectal Adenocarcinoma, HT29.	Irradiated with white light at 681 nm in 110 W/cm^2^ and fluence 20 J/cm^2^ for 18 to 24 h.	Pheophorbide a-poly(N-isopropylacrylamide) nanohydrogel with reasonable biocompatibility and acceptable photocytotoxicity in the low μM range.	[120]
2	Alginate-Based Microcapsules with a Molecule Recognition Linker and Photosensitizer for the Combined Cancer Treatment	30 μM of Ca-ALG (HB-lipid), and Ca-ALG-DOX-(HB-lipid) hydrogels on Immortal cells, HeLa.	Irradiated with blue light at 488 nm in 0.5 W/cm^2^, and fluence 30 J/cm^2^ for 36 h.	Ca-ALG (HB-lipid) and Ca-ALG-DOX-(HB-lipid) hydrogels are the co-delivery carriers with high efficiency in treating PDT against Immortal cells (HeLa).	[121]
3	Curcumin and silver nanoparticles carried out from polysaccharide-based hydrogels improved the photodynamic properties of curcumin through metal-enhanced singlet oxygen effect	91.5 μM of CHT/CS/CUR-AgNPs hydrogel on Human Colon Cancer cells, Caco-2.	Irradiated with green light at 525 nm in 420 W/cm^2^, and fluence 50 J/cm^2^ for 24 h.	PDT selective illumination led to the inhibition of Human Colon Cancer cells (Caco-2) by the CHT/CS/CU R-AgNPs hydrogel, and CUR can work as a diagnostic fluorescence probe in this system.	[122]

### 3.1. Resveratrol Combined with Hydrogel

Resveratrol has limitations in clinical studies, including poor bioavailability, low water solubility, and chemical instability in neutral and alkaline environments [123]. Hydrogel is a special material, and Resveratrol overcomes the above issue with the help of this. A hydrogel has several properties, such as high biocompatibility, drug protection, spatiotemporal control of the drug release, and physicochemical tailorability. Some examples are summarized in Table 7.

#### Mechanism of Combination

The mechanism for the combination of Resveratrol with hydrogel against bacteria and cancer is based on the antimicrobial peptide, antimicrobial agents, antibiotics, and polysaccharide, e.g., chitosan or cyclodextrin, e.g., (i) incorporation of Resveratrol–hydroxypropyl-β-cyclodextrin complexes into hydrogel formulation for wound treatment against *Staphylococcus aureus*, *Escherichia coli*, and *Candida albicans* (Figure 10) [126], and (ii) chitosan-based injectable in situ forming hydrogels containing dopamine-reduced graphene oxide and Resveratrol for breast cancer chemo-photothermal therapy (Figure 11) [130].

Based on the above information, it is possible to have a photodynamic action of Resveratrol with hydrogel against bacteria and cancer since the hydrogel is a three-dimensional (3D) network structure with crosslinked polymer chains that forms the ionic or covalent bond for linking one polymer chain to another, which provides high biocompatibility, drug protection, spatiotemporal control of the drug release, and physicochemical tailorability.

Resveratrol has antibacterial and anticancer effects. It affects the changes in cell morphology and DNA contents [133]. It is also able to inhibit the carcinogenesis stages, including initiation, promotion, and progression [134,135,136]. These are the reasons for using Resveratrol as a PS or adjuvant.

There is potential for the photodynamic activity of Resveratrol for treating various types of bacteria and cancers, with the particular advantage of causing minimal toxic side effects. This may increase the sensitivity of some cancer cells against chemotherapy drugs and overcome one or more of the body’s mechanisms, which eliminate the side effects of chemotherapy, such as anorexia, fatigue, depression, nerve pain, and cognitive impairment [137].

Regarding the photodynamic activity of Resveratrol, it is much better combined with the hydrogel. This eliminates the clinical application problems of Resveratrol, such as poor bioavailability, low water solubility, and chemical instability in neutral and alkaline environments. It greatly enhances the effectiveness of PDT against bacteria and cancer.

Moreover, hydrogel has started to be used with nanoparticles in recent research. Yang et al. reported that Resveratrol induced apoptosis and inhibited tumor growth after being coated with gold nanoflowers. It also enriched the tumor sites and identified tumor sites by computed tomography (CT) [138]. Xiang et al. identified Resveratrol-loaded dual-function titanium disulfide nanosheets for photothermal-triggered tumor chemotherapy with no remarkable tissue toxicity. Titanium disulfide nanosheets with Resveratrol can target and accumulate in mitochondria when triggered by near-infrared light. It induces the upregulation of key intrinsic apoptotic factors such as cytochrome c and initiates the caspase cascade to achieve the chemotherapeutic effect [139]. Thus, hydrogel nanoparticles or nanocomposites should be used in further investigation of Resveratrol with PDT in the treatment of bacteria and cancer.

However, there is some toxicity, and adverse effects were reported for the consumption of Resveratrol. Therefore, extensive studies on long-term effects and the in vivo adverse effects of Resveratrol supplementation in humans are needed [140]. These include concerns about the dosage of Resveratrol and duration time for humans if it is used as a PS or adjuvant.

## 4. Conclusions

Resveratrol combined with hydrogel is suitable for PDT treatment to fight against bacteria and cancer. Hydrogel consists of three-dimensional (3D) network structures with natural, synthetic, or semi-synthetic polymers through physical or chemical crosslinked methods. It has several competitive advantages, including good biocompatibility and low cytotoxicity, enhanced antitumor effect, reduced toxic and side effects, as well as maintaining a high concentration and archive the selective of a drug during the PDT process.

Resveratrol is a natural polyphenol stilbene structure containing two isomeric *cis*- and *trans*-forms. *Trans*-Resveratrol is dominant and more stable with significant photodynamic activity, which acts as a PS to generate ROS during the PDT process. However, *Trans*-Resveratrol has some limitations, with poor water solubility the major issue for biological application. Thus, they are compatible and reinforce each other to increase the effectiveness of PDT against bacteria and cancer.

However, more work is required, especially for the cytotoxicity safety assessment of the human body. The selection of hydrogel and Resveratrol acting as PS or adjuvant to enhance the effectiveness of PDT is another important milestone in the future.

## 5. Future Aspects

How do we enhance the effectiveness and therapeutic effect of PDT with natural products against bacteria and cancer in the future? There are some strategies to improve photodynamic therapy efficacy, such as making good use of non-reactive oxygen carriers (microbubbles/nano-bubbles, hemoglobin, and perfluorocarbon), reactive oxygen carriers (PDT dependent/independent materials), and regulating the microenvironment (blood perfusion, target mitochondria, moderate the level of Hypoxia-inducible factor 1, and hypoxia-activated therapy) [141]. Nanotechnology is another useful approach including the applications of liposomes, nanoparticles, and quantum dots [142]. In 2022, Li et al. developed the nanocomposite AuNS@ZrTCPP-GA (AZG), containing gambogic acid (GA), heat-shock protein 90 (HSP90) inhibitor, and the gold nanostars (AuNS) coated with PEGylated liposome (LP) to increase the stability and biocompatibility for enhancing the anticancer effect of PDT [143].

Meanwhile, it is suggested that the combination of photodynamic (PD) and photothermal (PT) therapies harness light to eliminate cancer cells with spatiotemporal precision by either generating ROS or increasing temperature. This addresses the limitations of the PDT/PTT modality and enhances treatment safety as well as efficacy. However, the complicated preclinical assessment of PDT/PTT combinations and possible rationale or guidelines to elucidate the mechanisms underlying PDT/PTT interactions are required for further investigations [144].

## Figures and Tables

**Figure 1 biomedicines-12-02095-f001:**
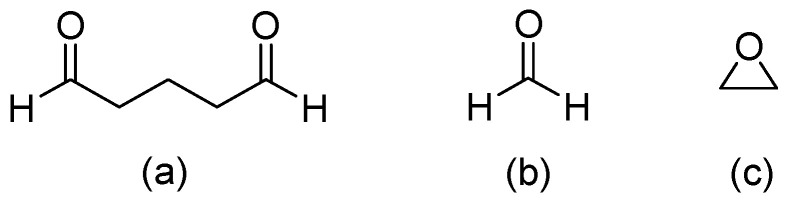
Crosslinking compounds of hydrogel: (**a**) Glutaraldehyde, (**b**) Formaldehyde, and (**c**) Epoxy.

**Figure 2 biomedicines-12-02095-f002:**
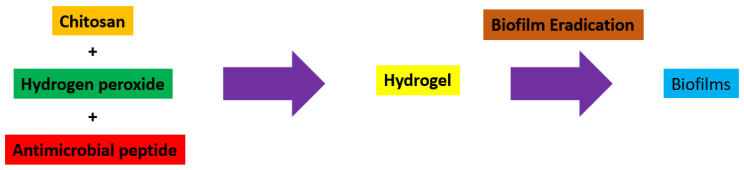
Synthetic diagram for the synthesis of a chitosan-based hydrogel with hydrogen peroxide of antimicrobial peptide against *Staphylococcus aureus*.

**Figure 3 biomedicines-12-02095-f003:**
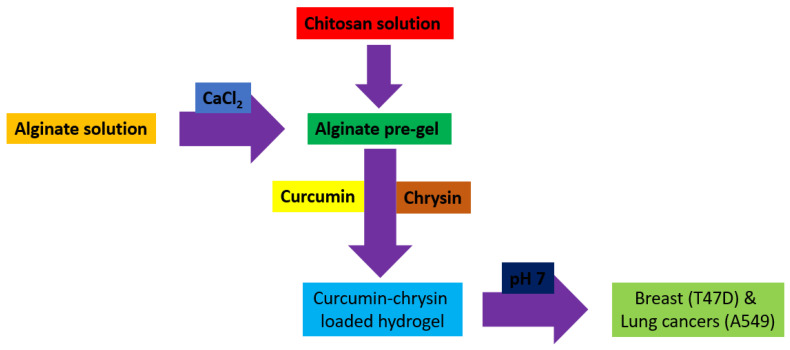
Synthetic diagram for the synthesis of a curcumin-chrysin-alginate-chitosan hydrogel against breast (T47D) and lung cancers (A549).

**Figure 4 biomedicines-12-02095-f004:**
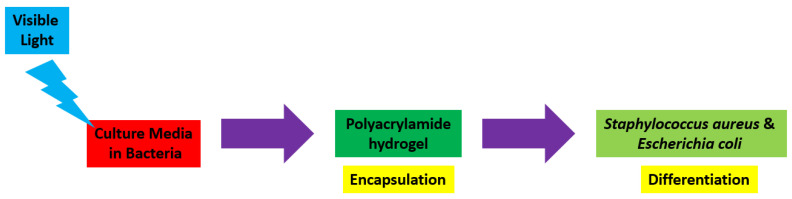
Synthetic diagram for the synthesis of a polyacrylamide hydrogel against *Staphylococcus aureus* and *Escherichia coli*.

**Figure 5 biomedicines-12-02095-f005:**
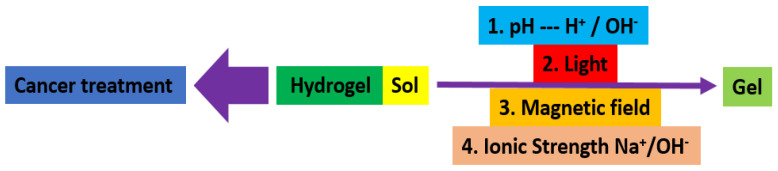
Synthetic diagram for the synthesis of a stimuli−responsive hydrogel for cancer treatment.

**Figure 6 biomedicines-12-02095-f006:**
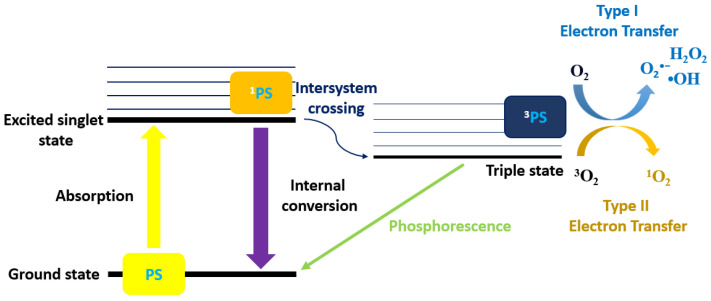
Schematic diagram of the PDT Types I and II mechanisms.

**Figure 7 biomedicines-12-02095-f007:**
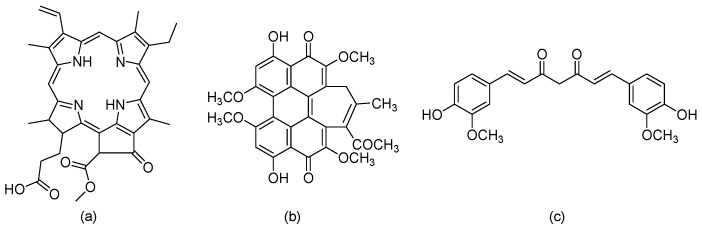
Chemical structures of (**a**) Pheophorbide a (Pa), (**b**) Hypocrellin B (HB), and (**c**) Curcumin (Cur).

**Figure 8 biomedicines-12-02095-f008:**
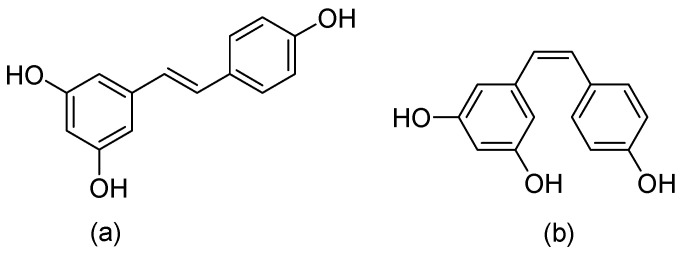
Chemical structures of (**a**) *trans*-Resveratrol, and (**b**) *cis*-Resveratrol.

**Figure 9 biomedicines-12-02095-f009:**
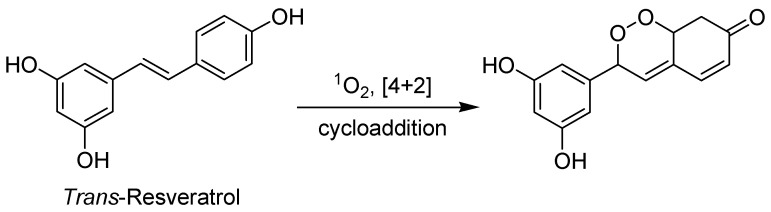
*Trans*-Resveratrol photooxidation of ergosterol via a [4+2] cycloaddition.

**Figure 10 biomedicines-12-02095-f010:**
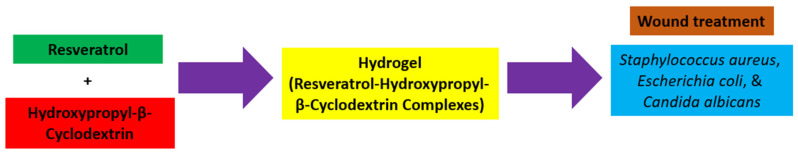
Synthetic diagram for the incorporation of Resveratrol–hydroxypropyl–β-cyclodextrin complexes into hydrogel formulation against bacteria.

**Figure 11 biomedicines-12-02095-f011:**
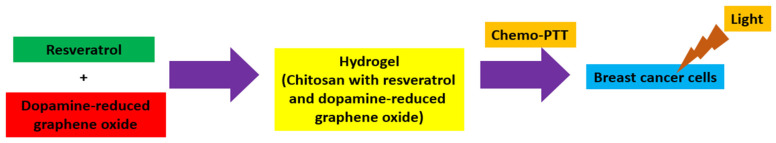
Synthetic diagram for the chitosan-based hydrogels containing dopamine-reduced graphene oxide and Resveratrol against breast cancer.

**Table 1 biomedicines-12-02095-t001:** Classification of hydrogel for the formation of crosslinking (modified from [10]).

**Crosslinking Methods**	Physical crosslinking	Hydrogen bonding
		Electrostatic interaction
		Van der Waals forces
		Host-guest interactions
		Crystallization
	Chemical crosslinking	Free radical reaction
		Carbodiimide chemistry
		Click chemistry
		Enzyme-mediated reaction
		Condensation polymerization

**Table 2 biomedicines-12-02095-t002:** Recent progress in PDT against bacteria and cancer.

Bacteria			
	Study	Consequences	References
1	Type-I photodynamic antimicrobial therapy: Principles, progress, and future perspectives	The fundamental principles of Type-I PDT were discussed, including its physicochemical properties and the generation of ROS, as well as explored several specific antimicrobial mechanisms utilized by Type-I PDT and summarized the recent applications of Type-I PDT in antimicrobial treatment.	[36]
2	Photodynamic therapy for the treatment of *Pseudomonas aeruginosa* infections: A scoping review	PDT was an effective adjunct to antimicrobial therapy against *Pseudomonas aeruginosa*, according to the usage of PS and the infection location, but the evidence was supported significantly by in vitro than the in vivo studies.	[37]
3	Antibacterial Photodynamic Therapy in the Near-Infrared Region with a Targeting Antimicrobial Peptide Connected to a π-Extended Porphyrin	Antimicrobial PDT upon irradiation at 720 nm for the conjugation consisted of an antimicrobial peptide linked to a π-extended porphyrin photosensitizer, which was at micromole concentration with a strong effect on both Gram-positive and Gram-negative bacteria.	[38]
**Cancer**			
	**Study**	**Consequences**	**References**
4	Recent Progress and Trends in X-ray-Induced Photodynamic Therapy with Low Radiation Doses	The concept of X-PDT and its relationships with radiodynamic therapy and radiotherapy, as well as the mechanism of X-ray absorption and conversion by scintillating materials, ROS evaluation for low dosage X-PDT, radiation side effects, and clinical concerns on X-ray radiation, were discussed.	[39]
5	Progress in Clinical Trials of Photodynamic Therapy for Solid Tumors and the Role of Nanomedicine	The latest clinical studies and preclinical in vivo studies on the use of PDT and its progress on nano-therapeutics as delivery tools for PS, which improved their cancer cellular uptake and their toxic properties	[40]
6	Trial watch: an update of clinical advances in photodynamic therapy and its immunoadjuvant properties for cancer treatment	Trial Watch provided recent clinical information on the immunomodulatory properties of PDT and ongoing clinical trials using PDT to treat cancer patients.	[41]

**Table 3 biomedicines-12-02095-t003:** Natural PS for PDT against bacteria and cancer.

Bacteria					
	Study	Natural Photosensitizer and Dosage	Usage of Light and Energy (J)	Consequences	Reference
1	Bactericidal Effect of Photodynamic Therapy Using Na-Pheophorbide a: Evaluation of Adequate Light Source	280 μM of Na-Pheophorbide a on *Staphylococcus aureus*.	Irradiated with red light at 670 nm in 300 W/cm^2^, and fluence 27 J/cm^2^ for 30 min.	PDT with Na-Pheophorbide a possessed a better anti-bactericidal function that was useful for the treatment of septic arthritis and soft tissue infection.	[54]
2	Hypocrellin B-Mediated Photodynamic Inactivation of Gram-Positive Antibiotic-Resistant Bacteria: An in vitro Study	100 μM of Hypocrellin B on *Staphylococcus aureus*, *Enterococcus faecalis*, *Streptococcus pneumonia*, *Escherichia coli*, and *Klebsiella pneumoniae*.	Irradiated with red light at 660 nm in 0.5 W/cm^2^, and fluence 72 J/cm^2^ for 30 min.	PDT with Hypocrellin B was effective in inactivating the Gram-positive bacteria, including *Staphylococcus aureus*, *Enterococcus faecalis*, and *Streptococcus pneumonia* bacteria.	[55]
3	Antimicrobial photodynamic therapy with curcumin on methicillin-resistant *Staphylococcus aureus* biofilm	80 μM of Curcumin on *Staphylococcus aureus*	Irradiated with blue light at 450 nm in 110 W/cm^2^, and fluence 50 J/cm^2^ for 455 s.	PDT with curcumin reduced the biofilm viability of *Staphylococcus aureus*, attesting to the efficiency of the therapy, which was internalized by bacterial cells even in biofilm aggregates,	[56]
**Cancer**					
	**Study**	**Natural Photosensitizer and Dosage**	**Usage of Light and Energy (J)**	**Consequences**	**Reference**
1	Pheophorbide a-Mediated Photodynamic Therapy Triggers HLA Class I-Restricted Antigen Presentation in Human Hepatocellular Carcinoma	0.35 μM of Pheophorbide a on Human Hepatocellular Carcinoma, HepG2.	Irradiated with red light at 670 nm in 70 W/cm^2^, and fluence 84 J/cm^2^ for 20 min.	PDT with Pheophorbide a triggered phagocytic capture by human macrophages, causing apoptosis and cancer immunity in the tumor host.	[48]
2	Effect of photodynamic therapy with Hypocrellin B on apoptosis, adhesion, and migration of cancer cells	2.5 μM of Hypocrellin B on Human ovarian cancer, HO-8910.	Irradiated with red light at 660 nm in 0.5 W/cm^2^, and fluence 72 J/cm^2^ for 5 h.	PDT with Hypocrellin B induced apoptosis and inhibited adhesion and migration of cancer cells in vitro.	[57]
3	Assessing the Effects of Curcumin and 450 nm Photodynamic Therapy on Oxidative Metabolism and Cell Cycle in Head and Neck Squamous Cell Carcinoma: An in vitro Study	0.1 to 10 μM of Curcumin on Human Head and Neck Squamous Cell Carcinoma, HNSCC.	Irradiated with blue light at 450 nm in 0.25 W/cm^2^, and fluence 60 J/cm^2^ for 1 h.	PDT with Curcumin increased oxidative damage, reduced cellular growth, and a cell cycle block in the G1 phase for Human Head and Neck Squamous Cell Carcinoma.	[58]

**Table 4 biomedicines-12-02095-t004:** Antibacterial and anticancer effects of Resveratrol.

Bacteria					
	Study	Models	Dosage and Time	Consequences	Reference
1	Resveratrol antibacterial activity against Escherichia coli is mediated by Z-ring formation inhibition via suppression of FtsZ expression	*lacZ*^+^ gene of an SOS-inducible sulA promoter on *Escherichia coli*.	456 μM of Resveratrol for 6 h.	Resveratrol increased DNA fragmentation and the expression level of SOS response-related genes in a dose-dependent manner, which inhibits bacterial cell growth by suppressing FtsZ expression and Z-ring formation.	[72]
2	Chemically Tuning Resveratrol for the Effective Killing of Gram-Positive Pathogens	*Bacillus cereus, Clostridium strains*, *Clostridioides difficile*, *Enterococcus faecalis*, *Streptococcus aureus*	40 to 160 μM of Resveratrol and its derivatives for 8 h.	Resveratrol and its derivatives with bactericidal activity against Gram-positive bacteria in the same low micromolar range of traditional antibiotics for disturbing the membrane permeability.	[73]
3	The Antibacterial and Antibiofilm Activities of Resveratrol on Gram-positive and Gram-negative Bacteria	*Staphylococcus aureus*, *Bacillus subtilis*, *Escherichia coli*, and *Pseudomonas aeruginosa*.	0.5 to 16 μM of Resveratrol for 24 h.	Resveratrol with antibacterial effect on Gram-positive and Gram-negative bacteria reduced the toxin production and inhibited biofilm formation.	[74]
4	Resveratrol enhances the antimicrobial effect of polymyxin B on Klebsiella pneumoniae and Escherichia coli isolates with polymyxin B resistance	6 strains of *Klebsiella pneumoniae* and 24 strains of *Escherichia coli*.	4 μM and 2 μM of polymyxin B for 6 strains of *Klebsiella pneumoniae* and 24 strains of *Escherichia coli* with an additional 32 to 128 μM Resveratrol for 24 h.	Resveratrol has increased the sensitivity of bacterial strains to polymyxin B, which is suitable for the treatment of bacterial infections.	[75]
5	Resveratrol Reverts Tolerance and Restores Susceptibility to Chlorhexidine and Benzalkonium in Gram-Negative Bacteria, Gram-Positive Bacteria, and Yeasts	*Enterococcus faecium*, *Staphylococcus aureus*, *Klebsiella pneumoniae*, *Acinetobacter baumannii*, *Pseudomonas aeruginosa* and *Enterobacter* spp.	32 to 256 μM of Resveratrol with 0.06 to 1024 μM of chlorhexidine and benzalkonium for 18 to 24 h.	Resveratrol reduced the dosage of chlorhexidine and benzalkonium, which was useful for biocides in several nosocomial pathogens.	[76]
6	Resveratrol-Induced Xenophagy Promotes Intracellular Bacteria Clearance in Intestinal Epithelial Cells and Macrophages	Life-threatening diseases, *Salmonella Typhimurium* and Crohn’s disease-associated Adherent-Invasive *Escherichia coli*.	10 μM of Resveratrol for 20 h	Resveratrol stimulated xenophagy and enhanced the clearance of two invasive bacteria, including *Salmonella Typhimurium* and *Escherichia coli*.	[77]
**Cancer**					
	**Study**	**Models**	**Dosage and Time**	**Consequences**	**Reference**
1	Resveratrol triggers autophagy-related apoptosis to inhibit the progression of colorectal cancer via inhibition of FOXQ1	Female BALB/c nude mice and C57BL/6 mice aged 6–8 weeks, or SW480-derived cancer cells.	Concentration between 50 and 80 μM for 48 h.	Resveratrol enhanced autophagy-related cell apoptosis in colorectal cancer cells through the SIRT1/FOXQ1/ATG16L pathway	[78]
2	Synergistic anticancer activity of Resveratrol in combination with docetaxel in prostate carcinoma cells	Prostate carcinoma LNCaP cells.	20 μM and 2 μM docetaxel for 24, 48, and 72 h.	Resveratrol improved the docetaxel therapy, targeting apoptosis and necroptosis simultaneously in the treatment of cancer	[79]
3	Metformin enhances anticancer properties of Resveratrol in MCF-7 breast cancer cells via induction of apoptosis, autophagy, and alteration in cell cycle distribution	Breast cancer cells, MCF-7.	2.5, 10, 25 mM of metformin, 70, 144, 287, 383, 575 μM of Resveratrol, and 20, 40 μM of cisplatin for 48 h.	Synergistic anticancer effects of metformin in a triple combination with cisplatin and Resveratrol were attributed to induction of autophagy-mediated cell death and apoptosis along cell cycle arrest	[80]
4	Resveratrol enhanced the anticancer effects of cisplatin on non-small cell lung cancer cell lines by inducing mitochondrial dysfunction and cell apoptosis	Human Lung cancer cells, H838 and H520.	40 μM and 55 μM of Resveratrol with 5 μM in H838 and H520 for 24 h.	Resveratrol exhibited its anticancer effects on non-small cell lung cancer H838 and H520 cell lines and enhanced the antitumor effects of cisplatin by regulating the mitochondrial apoptotic pathway.	[81]
5	Resveratrol Modulates the Redox-status and Cytotoxicity of Anticancer Drugs by Sensitizing Leukemic Lymphocytes and Protecting Normal Lymphocytes	Human Leukemic cells, U937 and HL-60.	12.5 μM of Resveratrol with 0.05 μM or 0.01 μM of barasertib, and 5 μM of everolimus.	In leukemic lymphocytes treated with barasertib and everolimus in the presence of Resveratrol, synergistic cytotoxicity was accompanied by strong induction of apoptosis, increased levels of hydroperoxides, and insignificant changes in protein-carbonyl products.	[82]
6	Efficacy of Resveratrol against breast cancer and hepatocellular carcinoma cell lines	Breast cancer cells MCF-7, and Hepatoblastoma cells, HepG2.	100 μM of Resveratrol h in MCF-7 and HepG2 for 24 h.	Resveratrol had a significant cytotoxic effect on MCF-7 and HepG2, which elevated caspase-3, caspase-8, caspase-9, Bax, p53, and p21 and reduced Bcl-2 and Bcl-xL mRNA levels in treating breast and liver cancers.	[83]
7	Apoptotic effects of Resveratrol, a grape polyphenol, on imatinib-sensitive and resistant K562 chronic myeloid leukemia cells	Chronic myeloid leukemia cells K562 and K562/IMA-3 cells.	85 μM and 122 μM of Resveratrol in K562 and K562/IMA-3 cells for 72 h.	Resveratrol with antiproliferative and apoptotic effects on K562 cells, as well as increased more 100-folds in K562/IMA-3 cells.	[84]
8	Synergistic anticancer effects of curcumin and Resveratrol in Hepa1-6 hepatocellular carcinoma cells	Hepatocellular carcinoma cells, Hepa1-6.	40 μM and 10 μM of Resveratrol and curcumin for 48 h.	Resveratrol and curcumin induced the apoptosis of Hepa1-6 cells through the caspase-3, -8, and -9 activation, upregulated intracellular ROS levels, downregulated X-Linked Inhibitor of Apoptosis Protein (XIAP) and surviving expression.	[85]
9	In vivo Anticancer Effects of Resveratrol Mediated by NK Cell Activation	Natural killer cells, NK.	20 μM of Resveratrol and 5 μM of IL-2 for 36 h.	Resveratrol activated NK cells and synergistically increased IFN-γ secretion and NK cell cytotoxicity with interleukin-2 (IL-2), which inhibited tumor growth and metastasis in mice.	[86]
10	Celastrol and Resveratrol Modulate SIRT Genes Expression and Exert Anticancer Activity in Colon Cancer Cells and Cancer Stem-like Cells	Metastatic colon cancer LoVo cells and cancer stem-like cells LoVo/DX.	1.0 to 5.0 μM of Resveratrol and 1.0 to 5.0 μM of celastrol for 24 h.	Resveratrol and celastrol exerted an antitumor activity against metastatic LoVo cells and cancer stem-like LoVo/DX cells, while Resveratrol with a greater effect on colon cancer cells and less aggressive forms of the disease than celastrol.	[87]

**Table 5 biomedicines-12-02095-t005:** Photodynamic action of Resveratrol against bacteria and cancer.

Bacteria					
	Study	Natural Photosensitizer and Dosage	Usage of Light and Energy (J)	Consequences	Reference
1	Photodynamic Therapy with Resveratrol and an Nd:YAG Laser for *Enterococcus faecalis* Elimination	357 μM of Resveratrol and Nd:YAG on *Enterococcus faecalis*.	Irradiated with red light at 635 nm in 3.5 W/cm^2^, and fluence 14 J/cm^2^ for 48 h.	PDT with Resveratrol and Nd:YAG as the PS with pigment was efficacious for *Enterococcus faecalis* elimination without inducing any toxic effects on osteoblasts.	[106]
2	Photoactivated Resveratrol against *Staphylococcus aureus* infection in mice	No report of the Resveratrol use on *Staphylococcus aureus*	Irradiated with blue light at 450–495 nm in 75 W/cm^2^, and fluence 54 J/cm^2^ for 24 h.	PDT with Resveratrol generated singlet oxygen effects on the immune system, triggering TNF-α and IL-17A production to the clearance of *Staphylococcus aureus*.	[107]
3	Photoactivated Resveratrol controls intradermal infection by *Staphylococcus aureus* in mice: a pilot study	2 μM of Resveratrol on *Staphylococcus aureus*	Irradiated with blue light at 450 nm in 75 W/cm^2^, and fluence 22.5 J/cm^2^ for 5 min.	PDT with Resveratrol induced the expression of myeloperoxidase, greater bacterial clearance, and infection control by IL-10 production.	[108]
**Cancer**					
	**Study**	**Natural Photosensitizer and Dosage**	**Usage of Light and Energy (J)**	**Consequences**	**Reference**
1	The comparative effects of Resveratrol and Curcumin in combination with photodynamic therapy	10 mg/kg body weight of Resveratrol and 50 mg/kg body weight of Curcumin dissolved in 0.5% of carboxymethyl cellulose through oral gavage for 7 days, respectively, on walker carcinosarcoma in 66 Wistar Male Rats.	Irradiated with laser light at 685 nm in 25 W/cm^2^, and fluence 50 J/cm^2^ for 15 min.	PDT with Resveratrol and Curcumin decreased oxidative stress, diminished the Cyclooxygenase-2 and Nitric oxide synthase 2 expressions, and increased cell death by positively influencing the necrotic rate and apoptotic index.	[109]
2	Resveratrol enhances the effects of ALA-PDT on skin squamous cells A431 through the p38/MAPK signaling pathway	0.5 mM of Aminolevulinic Acid as PS and 58 μM of Resveratrol is an adjuvant on Skin Squamous Cells, A431.	Irradiated with red light at 635 nm in 50 W/cm^2^, and fluence 37 J/cm^2^ for 3 h.	Resveratrol enhanced the effect of ALA-PDT against skin squamous cells A431 through the p38/MAPK pathway.	[110]

**Table 7 biomedicines-12-02095-t007:** Hydrogel with Resveratrol against bacteria and cancers.

Bacteria					
	Study	Models	Dosage and Time	Consequences	Reference
1	Resveratrol-Loaded Hydrogel Contact Lenses with Antioxidant and Antibiofilm Performance	*Pseudomonas aeruginosa*, and *Staphylococcus aureus*	100 and 200 μM of Resveratrol for 24 h.	Resveratrol released from the hydrogels readily accumulated in tissues and was effective against *Pseudomonas aeruginosa*, and *Staphylococcus aureus*.	[124]
2	Liposomes-In-Hydrogel Delivery System Enhances the Potential of Resveratrol in Combating Vaginal Chlamydia Infection	*Chlamydia trachomatis*	1.5 and 3 μM of Resveratrol for 48 h.	The anti-chlamydial effect of RES was enhanced when incorporated into a liposomes-in-hydrogel delivery system, which was a promising option for the localized treatment of *Chlamydia trachomatis* infection.	[125]
3	Incorporation of Resveratrol–Hydroxypropyl-β-Cyclodextrin Complexes into Hydrogel Formulation for Wound Treatment	*Staphylococcus aureus*, *Escherichia coli*, and *Candida albicans*	0.35 and 0.175 μM of Resveratrol–Hydroxypropyl-β-Cyclodextrin Complexes.	Resveratrol–Hydroxypropyl-β-Cyclodextrin Complexes were included in Pluronic hydrogel, which provided efficient drug release and appropriate viscoelastic properties for wound treatment.	[126]
4	Breathable hydrogel dressings containing natural antioxidants for the management of skin disorders	*Staphylococcus aureus*	15 and 30 μM of Resveratrol.	The developed hydrogel patch was breathable and able to maintain excellent mechanical properties with Resveratrol for 72 h against bacterial growth.	[127]
5	Resveratrol therapeutics combines both antimicrobial and immunomodulatory properties against respiratory infection by nontypeable Haemophilus influenzae	*Escherichia coli*, and *Bacillus subtilis*	112.5 and 56.25 μM of Resveratrol.	Resveratrol was therapeutic in targeting chronic obstructive pulmonary disease airway infections.	[128]
**Cancer**					
	**Study**	**Models**	**Dosage and Time**	**Consequences**	**Reference**
1	Injectable click-crosslinked hydrogel containing Resveratrol to improve the therapeutic effect in triple-negative breast cancer	Breast cancer cells, MDA-MB-231.	0.5 μM of Resveratrol and Resveratrol with hyaluronic acid for 24 h.	Resveratrol with hyaluronic acid significantly reduced negative tumor growth rates coupled with large apoptotic cells and limited angiogenesis in tumors.	[129]
2	Chitosan-based injectable in situ forming hydrogels containing dopamine-reduced graphene oxide and Resveratrol for breast cancer chemo-photothermal therapy	Breast cancer cells, MCF-7.	66 μM of Dopamine-reduced graphene oxide with Resveratrol for 24 h.	Resveratrol-formulated hydrogels displayed injectability and in situ gelation, as well as suitable physicochemical properties and good cytocompatibility, which was an enormous potential for the chemo-photothermal therapy of breast cancer cells.	[130]
3	Thermosensitive Hydrogels Loaded with Resveratrol Nanoemulsion: Formulation Optimization by Central Composite Design and Evaluation in MCF-7 Human Breast Cancer Cell Lines.	Breast cancer cells, MCF-7.	25 μM of Resveratrol with hydrogel for 6 h.	The developed Resveratrol with hydrogel was an effective delivery of breast cancer, and the in vitro release profile demonstrated a release rate of 80%.	[131]
4	Delivery of Resveratrol, a Red Wine Polyphenol, from Solutions and Hydrogels via the Skin	Female nude mice (ca. 8 weeks old) on skin erythema.	0.6 mL aliquot of Resveratrol hydrogel for 24 h.	Resveratrol with the hydrogel caused no stratum corneum disruption or skin erythema, and it was a therapeutic skin route of administration.	[132]

## Data Availability

Data are contained within the article.

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
