# Peer review of "Potential of Resveratrol to Combine with Hydrogel for Photodynamic Therapy against Bacteria and Cancer—A Review"

_biomedicines, 2024, doi:10.3390/biomedicines12092095_

Round 1
Reviewer 1 Report
Comments and Suggestions for Authors
The authors introduce the issues of antimicrobial and antibacterial activity, as well as new treatments based on photodynamic therapies. The authors mention that resverator is a compound extracted from Chinese plants. However, this compound can be extracted from plants growing in many other parts of the globe.
Hydrogels are presented too generally. Some examples of hydrogels should be mentioned.
The authors should also discuss what effect PDT has on the structure of hydrogels.
Author Response
Thanks for your comments. Enclosed is the pdf file.
Reviewer 1
The authors introduce the issues of antimicrobial and antibacterial activity, as well as new treatments based on photodynamic therapies.
Comment 1
The authors mention that resveratrol is a compound extracted from Chinese plants. However, this compound can be extracted from plants growing in many other parts of the globe.
Response 1
Thanks for your comment. It has been already mentioned that Resveratrol is isolated from the root of Veratrum grandiflorum [48], while darakchasava, or manakka for medicinal purposes [49] in the manuscript, although this plant growing in many other parts of the globe.
Comment 2
Hydrogels are presented too generally. Some examples of hydrogels should be mentioned.
Response 2
“1.2. Recent Examples of Antibacterial and Anti-cancer Applications” part will be added to the manuscript.
Hydrogels can be naturally derived from alginate, and chitosan, or synthetic modification from polyacrylamide. Fasiku et al reported a chitosan-based hydrogel for dual delivery with hydrogen peroxide of antimicrobial peptide against bacterial methicillin-resistant staphylococcus aureus biofilm-infected wounds that were prepared through the Michael addition technique (Figure 2) [22]. In 2022, Abbasalizadeh et al developed a curcumin-chrysin-alginate-chitosan hydrogel that was prepared through the ionic gelation mechanism utilizing CaCl2 to treat breast (T47D) and lung cancers (A549) (Figure 3) [23]. Lu et al identified the polyacrylamides hydrogel causing Staphylococcus aureus, and Escherichia coli differentiation upon visible light irradiation (Figure 4) [24]. Andrade et al indicated that stimuli-responsive hydrogel was able to change its physical state from liquid to gel according to external factors such as temperature, pH, light, ionic strength, and magnetic field for cancer treatment (Figure 5) [25].
Figure 2. Synthetic diagram for the synthesis of a chitosan-based hydrogel with hydrogen peroxide of antimicrobial peptide against staphylococcus aureus.
Figure 3. Synthetic diagram for the synthesis of a curcumin-chrysin-alginate-chitosan hydrogel against breast (T47D) and lung cancers (A549).
Figure 4. Synthetic diagram for the synthesis of a polyacrylamide hydrogel against Staphylococcus aureus, and Escherichia coli.
Figure 5. Synthetic diagram for the synthesis of a stimuli-responsive hydrogel for cancer treatment.
- Fasiku, V.O.; Omolo, C.A.; Devnarain, N.; Ibrahim, U.H.; Rambharose, S.; Faya, M.; Mocktar, C.; Singh, S.D.; Govender, T. Chitosan-Based Hydrogel for the Dual Delivery of Antimicrobial Agents Against Bacterial Methicillin-Resistant Staphylococcus aureusBiofilm-Infected Wounds. ACS Omega. 2021, 6, 21994-22010.
- Abbasalizadeh, F.; Alizadeh, E.; Bagher Fazljou, S.M.; Torbati, M.; Akbarzadeh, A. Anticancer Effect of Alginate-chitosan Hydrogel Loaded with Curcumin and Chrysin on Lung and Breast Cancer Cell Lines. Curr Drug Deliv. 2022, 19, 600-613.
- Lu, H.; Huang, Y.; Lv, F.; Liu, L.; Ma, Y.; Wang, S. Living Bacteria-Mediated Aerobic Photoinduced Radical Polymerization for in Situ Bacterial Encapsulation and Differentiation. CCS Chemistry. 2021, 3, 1296-1305.
- Andrade, F.; Roca-Melendres, M.M.; Durán-Lara, E.F.; Rafael, D.; Schwartz, S. Jr. Stimuli-Responsive Hydrogels for Cancer Treatment: The Role of pH, Light, Ionic Strength and Magnetic Field. Cancers (Basel). 2021, 9, 13, 1164.
Comment 3
The authors should also discuss what effect PDT has on the structure of hydrogels.
Response 3
This information is added to the “3. Photodynamic Action of Hydrogel”.
PDT can lead to additional crosslinking within the hydrogel network, which reacts with polymer chains, altering the mechanical properties and swelling behavior of the hydrogel. The incorporation of photosensitizers into hydrogels greatly enhances their localized concentration and biocompatibility. It prolongs their residence time to generate ROS against bacteria and cancer during the PDT process [121]. However, PDT also may cause the degradation of hydrogel to release a drug (e.g. Resveratrol) since ROS breaks down polymer chains at the same time, resulting in fragmentation and affecting overall stability. The drug delivery system is the most important point for the hydrogel applications on antibacterial and anticancer functions [122].
- Gan, S.; Wu, Y.; Zhang, X.; Zheng, Z.; Zhang, M.; Long, L.; Liao, J.; Chen, W. Recent Advances in Hydrogel-Based Phototherapy for Tumor Treatment. Gels. 2023, 9, 286.
- Zhang, Y.; Tian, S.; Huang, L.; Li, Y.; Lu, Y.; Li, H.; Chen, G.; Meng, F.; Liu, G.L.; Yang, X.; Tu, J.; Sun, C.; Luo, L. Reactive oxygen species-responsive and Raman-traceable hydrogel combining photodynamic and immune therapy for postsurgical cancer treatment. Nat Commun. 2022, 13, 4553.

Reviewer 2 Report
Comments and Suggestions for Authors
A good review. I suggest following revisions.
1) Author should explain in details - why the hydrogels are using in PDT?
2) In Section 1.2 , Author should provide classification of drug delivery system in a table.
3) Section 2, Should include table of recent progress in PDT.
4) 3.1 section should include mechanism of combination.
5) Conclusion should be more scientific and details.
Author Response
Thanks for your comments. Enclosed is the PDF file.
Reviewer 2
A good review. I suggest the following revisions.
Comment 1
Author should explain in details - why the hydrogels are using in PDT?
This information is added to the “1.4. Reasons for Hydrogel Use in PDT”.
Response 1
In general, hydrogel has several competitive advantages, including good biocompatibility and low cytotoxicity, enhanced antitumor effect, reduced toxic and side effects, as well as maintaining a high concentration and archive the selective of a drug during the PDT process. These characteristics of hydrogel overcome the limitations of PDT and enhance its effectiveness, which has been discussed in drug delivery systems (1.3). Previously studied for a novel hydrogel shell on cancer cells that were prepared through in situ photopolymerization of polyethyleneglycol diacrylate (PEGDA) using methylene blue (MB) sensitized mesoporous titanium oxide (TiO2) nanocrystal for enhancing the effectiveness of PDT. TiO2 was the PS and activated the formation of hydrogel, which protected the MB and acted as a significant photosensitive additive to improve the treatment of PDT. MB would be eliminated and inactivated after undergoing the PDT process [32]. Thus, there is a relationship between the principle of PDT and hydrogel.
- Chang, G.; Zhang, H.; Li, S.; Huang, F.; Shen, Y.; Xie, A. Effective photodynamic therapy of polymer hydrogel on tumor cells prepared using methylene blue sensitized mesoporous titania nanocrystal. Mater Sci Eng C Mater Biol Appl. 2019, 99, 1392-1398.
Comment 2
In Section 1.3., Author should provide classification of drug delivery system in a table.
Response 2
A table is attached to section 1.3. for the classification of the drug delivery system.
Table 1. Classification of hydrogel for the formation of cross-linking (modified from [31]).
Cross-linking methods |
Physical cross-linking |
Hydrogen bonding |
Electrostatic interaction |
||
Van der Waals forces |
||
Host-guest interactions |
||
Crystallization |
||
Chemical cross-linking |
Free radical reaction |
|
Carbodiimide chemistry |
||
Click chemistry |
||
Enzyme-mediated reaction |
||
Condensation polymerization |
- Liu, B.; Chen, K. Advances in Hydrogel-Based Drug Delivery Systems. Gels. 2024, 10, 262.
Comment 3
Section 2, Should include table of recent progress in PDT.
Response 3
A table is enclosed in section 2, Principle of PDT.
Table 2. Recent progress in PDT against bacteria and cancer.
Bacteria |
|||
|
Study |
Consequences |
References |
1 |
Type I photodynamic antimicrobial therapy: Principles, progress, and future perspectives |
The fundamental principles of type-I PDT were discussed including its physicochemical properties, and the generation of ROS, as well as explored several specific antimicrobial mechanisms utilized by type-I PDT and summarized the recent applications of type-I PDT in antimicrobial treatment. |
[37] |
2 |
Photodynamic therapy for the treatment of Pseudomonas aeruginosa infections: A scoping review |
PDT was an effective adjunct to antimicrobial therapy against Pseudomonas aeruginosa, according to the usage of PS and the infection location, but the evidence was supported significantly by in vitro than the in vivo studies. |
[38] |
3 |
Antibacterial Photodynamic Therapy in the Near-Infrared Region with a Targeting Antimicrobial Peptide Connected to a π-Extended Porphyrin |
Antimicrobial PDT upon irradiation at 720 nm for the conjugation consisted of an antimicrobial peptide linked to a π-extended porphyrin photosensitizer, which was at micromole concentration with a strong effect on both Gram-positive and Gram-negative bacteria. |
[39] |
Cancer |
|||
|
Study |
Consequences |
References |
4 |
Recent Progress and Trends in X-ray-Induced Photodynamic Therapy with Low Radiation Doses |
The concept of X-PDT and its relationships with radiodynamic therapy and radiotherapy, as well as the mechanism of X-ray absorption and conversion by scintillating materials, ROS evaluation for low dosage X-PDT, radiation side effects, and clinical concerns on X-ray radiation were discussed. |
[40] |
5 |
Progress in Clinical Trials of Photodynamic Therapy for Solid Tumors and the Role of Nanomedicine |
The latest clinical studies and pre-clinical in vivo studies on the use of PDT and its progress on nano-therapeutics as delivery tools for PS, which improved their cancer cellular uptake and their toxic properties |
[41] |
6 |
Trial watch: an update of clinical advances in photodynamic therapy and its immunoadjuvant properties for cancer treatment |
Trial Watch provided recent clinical information on the immunomodulatory properties of PDT and ongoing clinical trials using PDT to treat cancer patients. |
[42] |
- Jiang, J.; Lv, X.; Cheng, H.; Yang, D.; Xu, W.; Hu, Y.; Song, Y.; Zeng, G. Type I photodynamic antimicrobial therapy: Principles, progress, and future perspectives. Acta Biomater. 2024, 177, 1-19.
- Yanten, N.; Vilches, S.; Palavecino, C.E. Photodynamic therapy for the treatment of Pseudomonas aeruginosa infections: A scoping review. Photodiagnosis Photodyn Ther. 2023, 44, 103803.
- Gourlot, C.; Gosset, A.; Glattard, E.; Aisenbrey, C.; Rangasamy, S.; Rabineau, M.; Ouk, T.S.; Sol, V.; Lavalle, P.; Gourlaouen, C.; Ventura, B.; Bechinger, B.; Heitz, V. Antibacterial Photodynamic Therapy in the Near-Infrared Region with a Targeting Antimicrobial Peptide Connected to a π-Extended Porphyrin. ACS Infect Dis. 2022, 8, 1509-1520.
- He, L.; Yu, X.; Li, W. Recent Progress and Trends in X-ray-Induced Photodynamic Therapy with Low Radiation Doses. ACS Nano. 2022, 16, 19691-19721.
- Alsaab, H.O.; Alghamdi, M.S.; Alotaibi, A.S.; Alzhrani, R.; Alwuthaynani, F.; Althobaiti, Y.S.; Almalki, A.H.; Sau, S.; Iyer, A.K. Progress in Clinical Trials of Photodynamic Therapy for Solid Tumors and the Role of Nanomedicine. Cancers (Basel). 2020, 12, 2793.
- Penetra, M.; Arnaut, L.G.; Gomes-da-Silva, L.C. Trial watch: an update of clinical advances in photodynamic therapy and its immunoadjuvant properties for cancer treatment. Oncoimmunology. 2023, 12, 2226535.
Comment 4
3.1. section should include mechanism of combination.
Response 4
This information is attached to 3.1.1. Mechanism of combination.
The mechanism for the combination of resveratrol with hydrogel against bacteria and cancer are based on the antimicrobial peptide, antimicrobial agents, antibiotics, and polysaccharide, e.g. chitosan, or cyclodextrin. It is similar to the (1.2). For example, (i) incorporation of resveratrol-hydroxypropyl-β-cyclodextrin complexes into hydrogel formulation for wound treatment against staphylococcus aureus, escherichia coli, and candida albicans (Figure 6)[134], (ii) chitosan-based injectable in situ forming hydrogels containing dopamine-reduced graphene oxide and resveratrol for breast cancer chemo-photothermal therapy (Figure 7) [139].
Figure 6. Synthetic diagram for the incorporation of resveratrol-hydroxypropyl-β-cyclodextrin complexes into hydrogel formulation against bacteria.
Figure 7. Synthetic diagram for the chitosan-based hydrogels containing dopamine-reduced graphene oxide and resveratrol against breast cancer.
Comment 5
Conclusion should be more scientific and details.
Response 5
Resveratrol combined with hydrogel is suitable for PDT treatment to fight against bacteria and cancer. “Hydrogel” consists of three-dimensional (3D) network structures with natural, synthetic, or semi-synthetic polymers through physical or chemical cross-linked methods. It has several competitive advantages, including good biocompatibility and low cytotoxicity, enhanced antitumor effect, reduced toxic and side effects, as well as maintaining a high concentration and archive the selective of a drug during the PDT process.
“Resveratrol” is a natural polyphenol stilbene structure containing two isomeric cis- and trans-forms. Trans-Resveratrol is dominant and more stable with significant photodynamic activity, which acts as a PS to generate ROS during the PDT process. However, Trans-Resveratrol has some limitations, poor water solubility is the major issue for biological application. Thus, they are compatible and reinforce each other to increase the effectiveness of PDT against bacteria and cancer.
However, much more work is required, especially for the cytotoxicity safety assessment of the human body. The selection of hydrogel and resveratrol acts as PS or adjuvant to enhance the effectiveness of PDT is another important milestone in the future.

Reviewer 3 Report
Comments and Suggestions for Authors
The current review consolidates the photodynamic therapy of resveratrol combine with Hydrogel. Though the present review is well written, the following major concerns to be resolved for the suitability of publication.
1. Introduction needs perfect continuation.
2. Author need to justify in the introduction section about the similar review has been published earlier.
3. Please remove the separate section of discussion in the review, and it may be merged.
4. Conclusion should be in details.
5. Future scope of the present review should be provided before the conclusion section.
6. Structural representation has to be provided, wherever requires. The present review is completely plain with the literature of the papers.
Author Response
Thanks for your comments. Enclosed is the pdf file.
Reviewer 3
The current review consolidates the photodynamic therapy of resveratrol combine with Hydrogel. Though the present review is well written, the following major concerns to be resolved for the suitability of publication.
Comment 1
Introduction needs perfect continuation.
Response 1
The comment is similar to review 1 and 2. Some parts are added to the introduction including “1.2. Examples of Antibacterial and Anti-cancer Applications”, and “1.4. Reasons for Hydrogel Use in PDT”.
Comment 2
Author need to justify in the introduction section about the similar review has been published earlier.
Response 2
This information is attached to the “1. Introduction”.
Recently, Gan and Liu have published similar reviews for the hydrogel-based phototherapy and drug delivery system [9-10], but these are not describing the natural product, “Resveratrol”.
- Gan, S.; Wu, Y.; Zhang, X.; Zheng, Z.; Zhang, M.; Long, L.; Liao, J.; Chen, W. Recent Advances in Hydrogel-Based Phototherapy for Tumor Treatment. Gels2023, 9, 286.
- Liu, B.; Chen, K. Advances in Hydrogel-Based Drug Delivery Systems. Gels. 2024, 10, 262.
Comment 3
Please remove the separate section of discussion in the review, and it may be merged.
Response 3
The subtitle “Discussion” has been removed, and merged to the previous part.
Comment 4
Conclusion should be in details.
Response 4
This comment is the same as reviewer 3. The conclusion has been re-written:
Resveratrol combined with hydrogel is suitable for PDT treatment to fight against bacteria and cancer. “Hydrogel” consists of three-dimensional (3D) network structures with natural, synthetic, or semi-synthetic polymers through physical or chemical cross-linked methods. It has several competitive advantages, including good biocompatibility and low cytotoxicity, enhanced antitumor effect, reduced toxic and side effects, as well as maintaining a high concentration and archive the selective of a drug during the PDT process.
“Resveratrol” is a natural polyphenol stilbene structure containing two isomeric cis- and trans-forms. Trans-Resveratrol is dominant and more stable with significant photodynamic activity, which acts as a PS to generate ROS during the PDT process. However, Trans-Resveratrol has some limitations, poor water solubility is the major issue for biological application. Thus, they are compatible and reinforce each other to increase the effectiveness of PDT against bacteria and cancer.
However, much more work is required, especially for the cytotoxicity safety assessment of the human body. The selection of hydrogel and resveratrol acts as PS or adjuvant to enhance the effectiveness of PDT is another important milestone in the future.
Comment 5
Future scope of the present review should be provided before the conclusion section.
Response 5
The subtitle part 5. Future Aspects have been provided.
How do enhance the effectiveness and therapeutic effect of PDT with natural products against bacteria and cancer in the future? There are some strategies to improve photodynamic therapy efficacy, such as making good use of non-reactive oxygen carriers (microbubbles/nano-bubbles, hemoglobin, and perfluorocarbon), reactive oxygen carriers (PDT dependent/independent materials), and regulating the microenvironment (blood perfusion, target mitochondria, moderate the level of Hypoxia-inducible factor 1, and hypoxia-activated therapy) [147]. Nanotechnology is another useful approach including the applications of liposomes, nanoparticles, and quantum dots [148]. In 2022, Li et al. developed the nanocomposite AuNS@ZrTCPP-GA (AZG) containing gambogic acid (GA), heat-shock protein 90 (HSP90) inhibitor, and the gold nanostars (AuNS) coated with PEGylated liposome (LP) to increase the stability and biocompatibility for enhancing the anticancer effect of PDT [149].
Meanwhile, it is suggested that the combination of photodynamic (PD) and photothermal (PT) therapies, harness light to eliminate cancer cells with spatiotemporal precision by either generating ROS or increasing temperature. This addresses the limitations of the PDT/PTT modality and enhances treatment safety, as well as efficacy. However, the complicated preclinical assessment of PDT/PTT combinations and possible rationale or guidelines to elucidate the mechanisms underlying PDT/PTT interactions are required for further investigations [150].
- Li, X.; Chen, L.; Huang, M.; Zeng, S.; Zheng, J.; Peng, S.; Wang, Y.; Cheng, H.; Li, S. Innovative strategies for photodynamic therapy against hypoxic tumor. Asian J Pharm Sci. 2023, 18, 100775.
- Olszowy, M.; Nowak-Perlak, M.; Woźniak, M. Current Strategies in Photodynamic Therapy (PDT) and Photodynamic Diagnostics (PDD) and the Future Potential of Nanotechnology in Cancer Treatment. Pharmaceutics. 2023, 15, 1712.
- Li, R.T.; Zhu, Y.D.; Li, W.Y.; Hou, Y.K.; Zou, Y.M.; Zhao, Y.H.; Zou, Q.; Zhang, W.H.; Chen, J.X. Synergistic photothermal-photodynamic-chemotherapy toward breast cancer based on a liposome-coated core-shell AuNS@NMOFs nanocomposite encapsulated with gambogic acid. J Nanobiotechnology. 2022, 20, 212.
- Overchuk, M.; Weersink, R.A.; Wilson, B.C.; Zheng, G. Photodynamic and Photothermal Therapies: Synergy Opportunities for Nanomedicine. ACS Nano. 2023, 17, 7979-8003.
Comment 6
Structural representation has to be provided, wherever requires. The present review is completely plain with the literature of the papers.
Response 6
Lots of schematic diagrams are updated. Based on reviewers 1 and 2 comments to correct and re-write the manuscript, the present review has a clear direction, which is different from the previous literature.

Round 2
Reviewer 2 Report
Comments and Suggestions for Authors
now can be accepted
Reviewer 3 Report
Comments and Suggestions for Authors
The author addresses all the comments satisfactorily. Hence, the revised version of the manuscript is now suitable for publication.